# A Research on Fault Diagnosis of a USV Thruster Based on PCA and Entropy

Ki-Beom Choo [1] , Hyunjoon Cho [2], Jung-Hyeun Park [2,3], Jiafeng Huang [2,3], Dongwook Jung [2,3], Jihyeong Lee [4,5], Sang-Ki Jeong [4], Jongsu Yoon [6], Jinhun Choo [6] and Hyeung-Sik Choi [2,*]

[1] Advanced-Intelligent Ship Research Division, Korea Research Institute of Ship & Ocean Engineering, Daejeon 34103, Republic of Korea
[2] Department of Mechanical Engineering, Korea Maritime & Ocean University, Busan 49112, Republic of Korea
[3] Interdisciplinary Major of Ocean Renewable Energy Engineering, Korea Maritime and Ocean University, Busan 49112, Republic of Korea
[4] Maritime ICT R&D Center, Korea Institute of Ocean Science and Technology, Busan 49111, Republic of Korea
[5] Ocean Science and Technology School, Korea Maritime and Ocean University, Busan 49111, Republic of Korea
[6] Autonomous Ship Technology Center, Korea Marine Equipment Research Institute, Busan 46754, Republic of Korea
* Correspondence: hchoi@kmou.ac.kr; Tel.: +82-010-5581-2971

**Abstract:** This study focuses on faults in the thrusters of unmanned surface vehicles, which are fatal to the integrity of their missions. As for the fault conditions, the breakage of the thruster blade and the entanglement of floating objects were selected, and a data-driven method was used to diagnose the faults. In the data-driven method, it is important to select the sensitive fault feature. In this study, vibration, current consumption, rotational speed and input voltage were selected as fault features. An experiment was conducted in an engineering water tank to obtain and analyze data on fault conditions to verify the validity of the selected features. In addition, a new fault diagnosis algorithm combining principal component analysis and Shannon entropy was applied for analyzing the correlations among fault features. This algorithm reduces the dimensionality of data while preserving their structure and characteristics, and diagnoses faults by quantifying entropy values. A fault is detected by comparing the entropy value and a predetermined threshold value, and is diagnosed by analyzing the entropy value and visualized 2D or 3D principal component results. Moreover, the fault diagnosis performance of the unmanned surface vehicle's thruster was verified by analyzing the results for each fault condition.

**Keywords:** USV; underwater thruster; fault diagnosis; PCA; Shannon entropy

## 1. Introduction

Unmanned platforms, such as unmanned surface vehicles (USVs) and unmanned underwater vehicles (UUVs), are actively developed for missions, such as marine environment surveys, monitoring, and management of offshore structures, using underwater thrusters as propulsion devices [1]. Research on improving reliability in autonomous navigation and operational safety has become more important as research on unmanned platforms has advanced [2]; thus, the need for fault diagnosis research has increased. However, unlike fault diagnosis studies for ground vehicles or UUVs [3–7], there has been a lack of research on fault diagnosis in USVs. In particular, a fault in the underwater thruster may cause critical losses to a mission; therefore, research on fault diagnosis for the underwater thrusters of unmanned platforms is crucial [8]. Moreover, recent studies lack consideration of actual experiments on platforms with thruster faults. It is crucial to conduct fault diagnosis for USV thrusters using actual trials that mimic real-world environments.

Previously reported fault diagnosis methods can be categorized into rule-based, model-based and data-driven approaches [6,9–11]. The rule-based method is applied to systems with simple structures, and fault diagnosis is performed by setting rules, such as thresholds.

The method has been employed in fault diagnosis research based on simulation by setting thresholds for the health level of thrusters and key equipment included in the USV [12]. The rule-based method requires expert knowledge for setting rules, and for each new type of fault additional restrictions need to be fixed, resulting in increased complexity in the system [13,14]. Moreover, the rule-based analysis method is inadequate in this research, since it is inappropriate to consider possible fault conditions in the future.

In the model-based method, a fault is diagnosed through the evaluation of residuals obtained by comparing the estimated values from a model of the entire system or subsystems and the measured values from data acquisition [15]. These model-based techniques have been applied in fault diagnosis research with modeling based on sensor data, such as position, velocity and attitude, and studies on fault-tolerant control [3,16,17]. USVs are operated close to the sea surface; therefore, their performance is significantly affected by environmental disturbances, such as wind, waves and currents, to which underwater thrusters are particularly vulnerable. Hydrodynamics induced by these environmental disturbances show highly nonlinear characteristics; hence, modeling such nonlinearity poses a challenge [18]. Therefore, a model-based approach was also considered inadequate for fault diagnosis of a USV thruster.

The data-driven approach entails diagnosing a fault by analyzing the pattern of acquired data. This method has no restrictions in terms of having to consider constraints of deriving a system model and has the advantage of being applicable to nonlinear systems [19–21]. Therefore, in comprehensive consideration of the various factors described above, the data-driven approach was selected as the optimal approach for fault diagnosis of USV thrusters.

Examples of the data-driven method include fault diagnosis studies on underwater thrusters using deep convolutional neural networks (DCNNs), one of the deep learning techniques, and t-Distributed Stochastic Neighbor Embedding (t-SNE) algorithms, one of the dimension reduction techniques, using multiple sensor data, such as hydrophone and current sensor data [22]. In addition, there has been research on thruster fault diagnosis based on discrete wave transforms and orthogonal fuzzy neighborhood discriminative analysis, one of the dimensionality reduction methods, using current and vibration data for the diagnosis of breakage conditions of thruster blades [23]. Furthermore, a study was performed using an actual USV to perform a water-tank experiment for the validation of selected fault features of a thruster [24]. In these previous studies on single-thruster fault diagnosis and in studies using actual USVs, analyses were performed on a frequency domain using wavelet or Fourier transform. In this study, we investigated a new fault diagnosis method based on multiple data correlation analysis for fault conditions and visualization.

The fault conditions chosen for this study were the breakage and entanglement of the thruster's blades, which are common faults caused by external factors [25]. By segmenting the faults and conducting an experiment with these cases, the type and degree of faults were identified and analyzed.

In the data-driven method, a fault is diagnosed based on data patterns, which makes it important to select appropriate fault features with sensitive responses to faults and which allow for the proper description of system characteristics. Fault features were selected with reference to previous studies in which the validity of fault features in the underwater thrusters of USVs were investigated [26]. To verify whether the selected fault features sensitively respond to fault conditions and properly reflect system states, experiments using an ocean engineering water tank were performed.

For the analysis of the correlation of data in the fault condition, different from other studies, the principal component analysis (PCA) algorithm, which allows the reduction of dimensionality and visualization while minimizing the loss of information in terms of structure and characteristics, was applied. Additionally, Shannon entropy was used for quantification to perform fault detection and diagnosis of USV thrusters.

The main contributions of our research can be described as follows:

(1)   In this study, data on vibration, current consumption, rotational speed and input voltage from the water-tank experiment and actual ship data were selected as fault features.

(2)   A new study of applying the fault diagnosis method for multiple thruster data correlation under fault conditions using a visualization scheme.

(3)   A fault diagnosis method is introduced that classifies fault types and is accessible for tuning in unstable-environment field practice.

(4)   The performance of the proposed fault diagnosis algorithm for an unmanned surface vehicle's thruster was verified by analyzing the results of each fault condition.

## 2. USV Thruster Fault Feature

### 2.1. USV Thruster Fault Condition

In this study, a fault was defined as the occurrence of at least one deviation from the expected behavior of the system that is not generally allowed. Fault detection is defined as the recognition of fault occurrence by the system or the user through symptoms using various methods. Fault diagnosis is defined as the process of detailed classification of fault types, such as the size of a fault, the location of its occurrence and the time taken for fault detection, from inputs on symptoms of the fault [26].

Figure 1 presents a schematic for the process of general fault diagnosis. First, fault features are extracted from acquired data. The fault features are examined to perform fault detection and recognize symptoms. Then, based on the identified symptoms, faults are classified for diagnosis. When the fault types are identified, the hazard of the fault is classified to facilitate the decision making of the user, such as the continuation of the mission, the repair of the fault or the return of the USV.

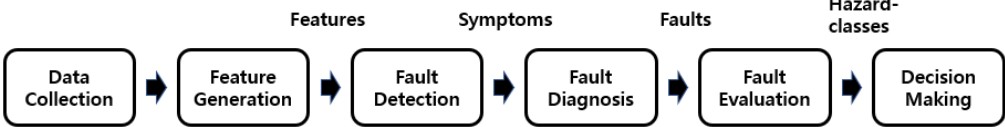

**Figure 1.** General procedure of fault diagnosis.

Among possible fault conditions caused by external factors of the system, the breakage of the thruster blade and entanglement of floating objects were selected as the faults to be detected and analyzed in this study.

Figure 2 outlines the illustration of the materials used for the reproduction of real-world faults in the study. A thin rope (~Ø6 mm), a thick rope (~Ø12 mm) and a net were used to reproduce conditions of entanglement of floating objects on the thruster blade. For reproduction of the breakage condition of the thruster blade, fault conditions were generated by breakage of one side of the two-leaf thruster blade by 1 cm, 2 cm and 3 cm, respectively. In this way, we aimed to study not only the diagnosis of the type of fault but also the severity of the fault by applying different sizes of the fault.

Before performing a fault diagnosis, appropriate selection of fault features for the acquired data is necessary. In particular, in the data-driven method, a fault is diagnosed based on the analysis of data patterns, which makes it crucial to select appropriate fault features according to the type of fault.

### 2.2. Analysis of Thruster Fault Features

To diagnose the selected faults (the breakage of the thruster blade and entanglement of floating objects), vibration, current consumption and rotational speed (revolutions per minute, RPM) of the thruster were selected as the fault features, in line with a previous study on the validation of fault feature selection for an underwater thruster [25]. In the proposed fault diagnosis method, correlations between data in fault conditions were analyzed to identify patterns, and faults were diagnosed by visualizing the identified patterns. Therefore, frequency-domain analysis, whereby the number of samples of data could be reduced, or filters with time delays were not applied; thus, the raw data-based method was used.

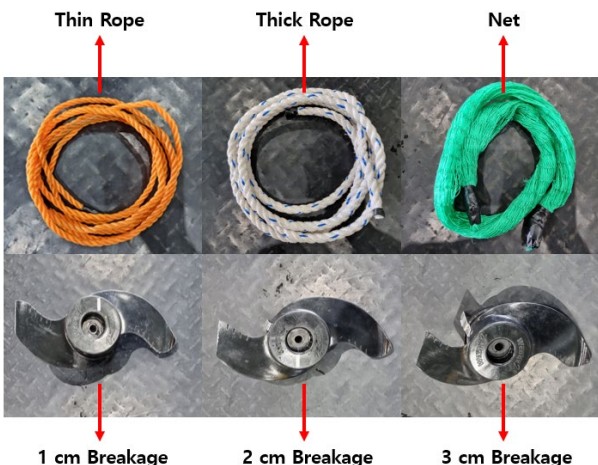

**Figure 2.** Materials for reproducing various fault conditions. Clockwise from top left: thin rope, thick rope, net, 3 cm breakage blade, 2 cm breakage blade and 1 cm breakage blade.

### 2.2.1. Vibration

When the thruster blade breaks or floating objects, such as ropes and nets, become entangled with the thruster, the geometric center and the center of mass of the thruster do not coincide with each other, resulting in mass unbalance. When a rotor operates in this state of unbalance, vibration is generated, as shown in Figure 3, and the formulas for the relationship are presented in Equations (1) and (2) [27].

$$F_c = em\omega^2 \tag{1}$$

$$X_{vib} = \frac{F_c}{K_{DS}} \tag{2}$$

where $F_c$ denotes the centrifugal force, $m$ represents the mass unbalance caused by breakage of the thruster blade and entanglement, $e$ indicates the distance between the center of mass and the geometric centerline, $\omega$ is the angular velocity, $X_{vib}$ is the vibration caused by the rotor unbalance and $K_{DS}$ represents the dynamic stiffness. Vibration is proportional to $m$ and $e$; therefore, if there is mass unbalance owing to the breakage of the thruster blade or entanglement of floating objects, $m$ and $e$ will increase, which, in turn, will increase vibration. Hence, vibration was selected as the fault feature.

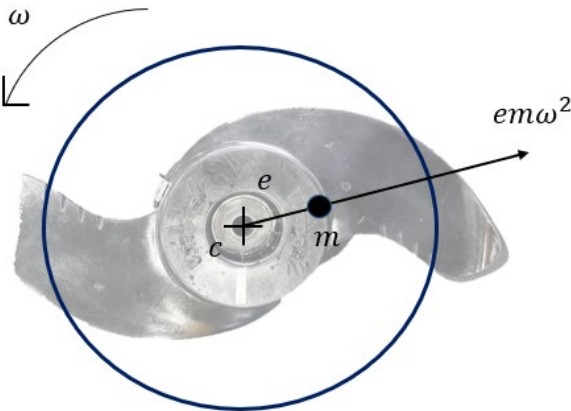

**Figure 3.** Vibration generated by mass unbalance.

### 2.2.2. Consumed Current and Rotation Speed

When blade breakage or the entanglement of floating objects with an underwater thruster happens, changes in torque occur due to the difference in the diameter or the

variation in the thruster blade area which pushes the fluid. A general equation representing the torque of the thruster is presented in Equation (3) [28].

$$Q = \rho K_Q n^2 D^5 \tag{3}$$

where $Q$ denotes the torque of the thruster, $\rho$ represents the fluid density, $K_Q$ is the torque coefficient, $n$ is the propeller revolutions per second and $D$ is the diameter of the propeller. Equation (4) represents a commonly known expression for the electrical torque of a DC motor [29].

$$Q = K_t I - J \frac{dn}{dt} \tag{4}$$

where $K_t$ indicates an electrical torque constant, $I$ is the current and $J$ is the moment of inertia. Assuming that there is no sudden change in the rotational speed of the thruster, Equation (4) can be simplified as follows:

$$Q = K_t I \tag{5}$$

Based on Equation (3), a torque equation for the thruster, and Equation (5), a simplified torque equation for a DC motor, the current equation can be expressed as follows:

$$I = \frac{\rho K_Q n^2 D^5}{K_t} \tag{6}$$

Assuming that the density, torque coefficient and electrical torque constant in Equation (6) are all constant, the consumed current $I$ is proportional to the square of the rotational speed $n$ and the fifth power of the diameter $D$ of the thruster. Therefore, because the diameter and the area where the thruster comes into contact with the fluid change according to the fault condition, it was conjectured that the consumed current and rotational speed would change; thus, these two variables were added to the fault features.

### 2.2.3. Input Voltage

In the case of the USV used in this study, the thrust system was composed of a battery, a motor drive and a thruster. Therefore, the battery voltage level and the thrust were directly connected. The input voltage was measured between the motor drive and the thruster; thus, the input voltage was additionally selected as the fault feature for reflecting this relationship.

## 3. Experimental Validation of Selected Fault Features

### 3.1. Configuration of a USV Fault Diagnosis System

In this study, a fault diagnosis system in a USV was designed and developed, as shown in Figure 4, to diagnose faults, specifically the breakage of the thruster blade and entanglement of floating objects. The USV consisted of a battery, a control system, two thrusters and a fault diagnosis system, all of which were placed in a rubber boat. The fault diagnosis system consisted of different sensors to measure the data of fault features and a data acquisition (DAQ) system for data collection and transmission.

Table 1 summarizes the types of sensors for acquiring data on the fault features and DAQ modules connected to each sensor.

### 3.2. Water-Tank Experiment

To validate the selected fault features, the normal, breakage and entanglement conditions of the thruster were reproduced, as shown in Figure 5. Data were obtained through experiments under normal conditions and the selected fault conditions in the ocean engineering water tank. In each experiment, data were acquired at a sampling rate of 1 kHz for 20 s. The battery voltage was lowered over time during the USV operation; therefore, a

total of 10 datasets were collected, five datasets each, under two different battery voltage conditions with a constant control input.

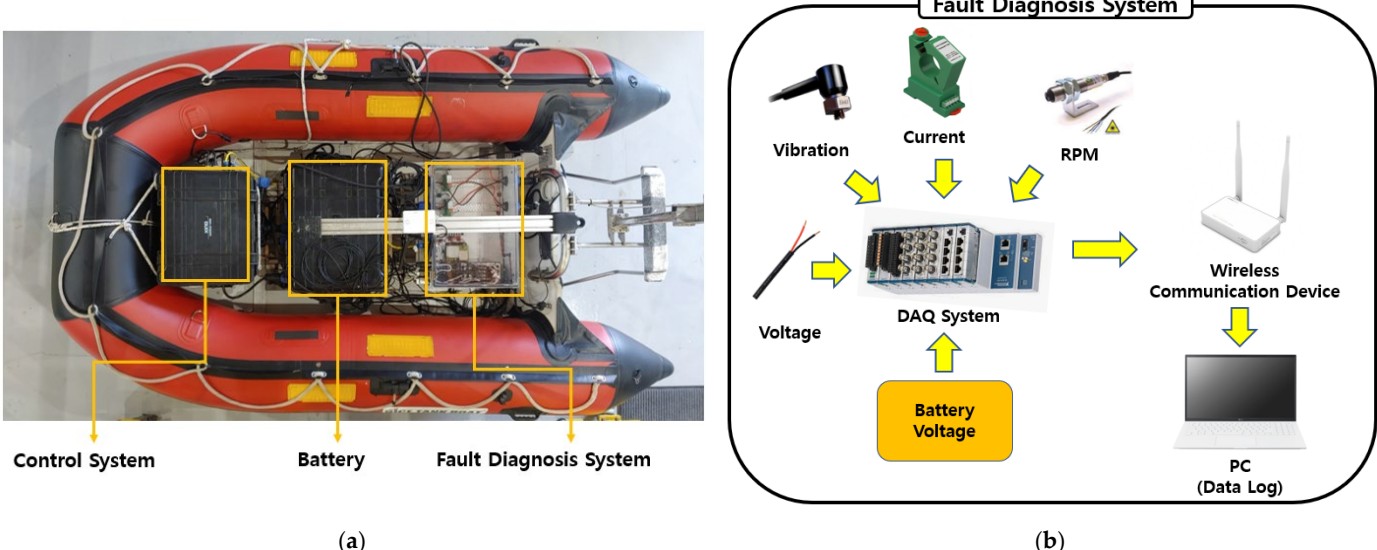

(**a**)　　　　　　　　　　　　　　　　　　　　　　　　　(**b**)

**Figure 4.** Configuration of the USV and fault diagnosis system. (**a**) The configuration of the USV; (**b**) The configuration of the fault diagnosis system.

**Table 1.** Specifications of the DAQ modules and sensors by type of acquired data.

| Data | DAQ Module | Sensor |
| --- | --- | --- |
| Thruster current | NI-9215 | CR Magnetics CR5210S-150 |
| Thruster input voltage | NI-9215 | - |
| Battery voltage | NI-9230 | - |
| Vibration | NI-9234 | PCB piezotronics 607A11 |
| Thruster RPM | NI-9423 | Monarch Instruments RLS |

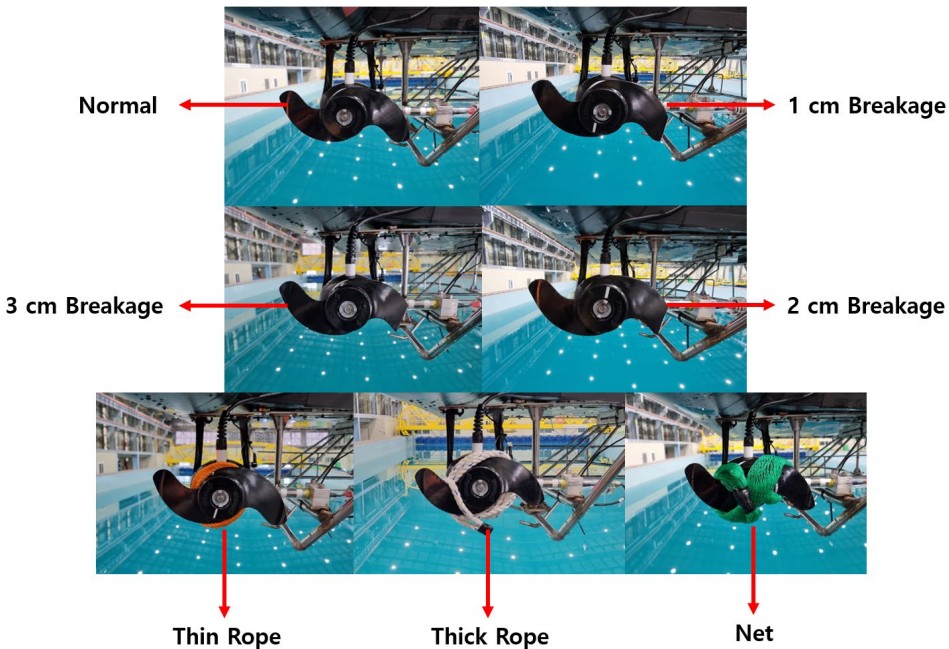

**Figure 5.** Fault conditions applied in experiments.

### 3.3. Results of the Water-Tank Experiment

The data obtained in the experiments under the normal condition and each fault condition of the thruster were classified into Case 1 and Case 2, according to the battery voltage, and the mean values and range of values for each data type are summarized in Table 2.

**Table 2.** Water-tank experiment results for normal and fault conditions.

| Fault Conditions | Case | Battery Voltage (V) | Input Voltage (V) | Consumed Current (A) | RPM | Vibration (g) |
|---|---|---|---|---|---|---|
| Normal | Case 1 | 28.10 | 18.87 | 16.46 | 821 | −0.1231~0.1327 |
| | Case 2 | 28.76 | 19.21 | 17.03 | 843 | −0.1359~0.1487 |
| 1 cm breakage | Case 1 | 28.06 | 18.58 | 16.78 | 811 | −0.2103~0.2062 |
| | Case 2 | 28.65 | 19.11 | 17.43 | 829 | −0.2041~0.1995 |
| 2 cm breakage | Case 1 | 28.42 | 18.58 | 16.11 | 822 | −0.3053~0.3010 |
| | Case 2 | 28.55 | 18.77 | 16.58 | 835 | −0.2665~0.2751 |
| 3 cm breakage | Case 1 | 28.02 | 18.53 | 14.71 | 842 | −0.2290~0.2685 |
| | Case 2 | 28.48 | 18.74 | 15.13 | 850 | −0.2463~0.2774 |
| Thin rope | Case 1 | 28.65 | 19.07 | 19.36 | 643 | −0.1738~0.1686 |
| | Case 2 | 28.33 | 18.71 | 22.66 | 752 | −0.2106~0.2307 |
| Thick rope | Case 1 | 28.49 | 18.50 | 27.36 | 706 | −0.5724~0.5654 |
| | Case 2 | 28.22 | 18.13 | 30.61 | 658 | −0.4944~0.4510 |
| Net | Case 1 | 28.43 | 18.36 | 28.23 | 692 | −0.2012~0.1951 |
| | Case 2 | 28.09 | 17.64 | 41.11 | 483 | −0.3771~0.3737 |

Using the data outlined in Table 2, the vibration data are illustrated as a graph in Figure 6; evidently, the vibration value under all the fault conditions was higher than that under the normal condition.

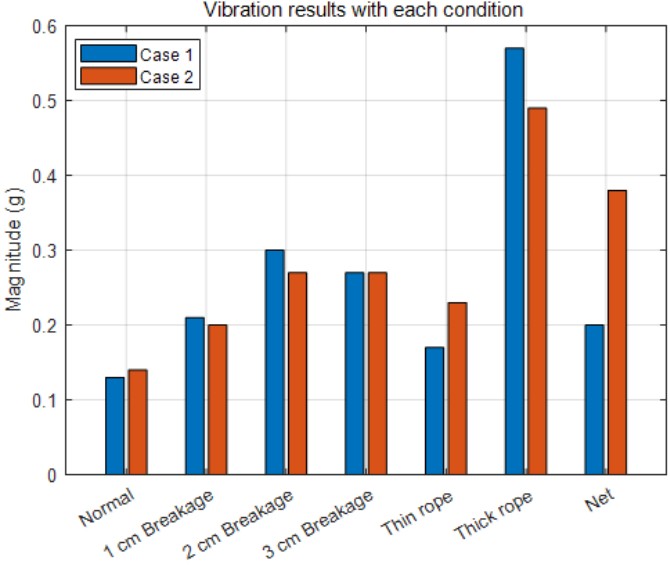

**Figure 6.** Vibration data for each condition.

In addition, the current and RPM data are illustrated in Figure 7. As shown by the data in Table 2 and Figure 7, under the fault condition of breakage, the consumed current decreased and the RPM increased according to the severity of the fault. Furthermore, under the fault condition of entanglement, the consumed current increased and the RPM decreased with the increasing severity of the fault.

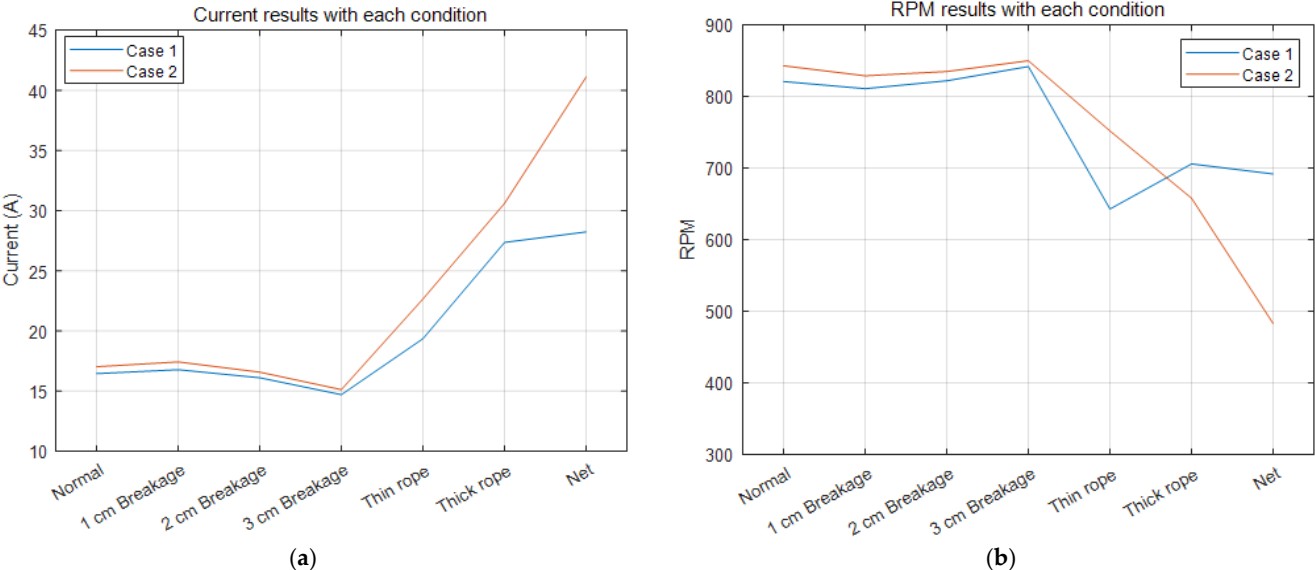

**Figure 7.** Current and RPM data for each condition. (**a**) Current data for each condition; (**b**) RPM data for each condition.

A more detailed analysis of the data change in Table 2 revealed that the magnitude of vibration generally increased under fault conditions. The increase was more pronounced under the conditions of 2 cm and 3 cm breakage compared to the condition of 1 cm breakage. There was no significant difference in the magnitude of vibration under the conditions of 2 cm and 3 cm breakage. However, under the breakage conditions, as the fault worsened, the consumed current slightly decreased, unlike the RPM, which increased slightly. This was attributed to a decrease in the diameter and area for pushing away the fluid.

Under the fault condition of entanglement, the general trend indicated that the consumed current increased and the rotation speed decreased. The fault was more severe in the thick rope compared to the thin rope. Compared to the net condition, it was observed that the current consumption and RPM increased noticeably in the entanglement condition.

In addition, the current value in the case of entanglement was higher when the net was involved than when the rope was involved because the net had a larger water-repellent surface area and was more flexible under the entanglement condition, resulting in a more significant change in the consumed current than in the vibration. Finally, it can be observed from the input voltage data in Table 2 that the input voltage also changed according to the battery voltage. Thus, through the analysis of changes in data according to the fault conditions, the selection of fault features in this study was validated.

Upon comparison of the results of each experiment, it was observed that the values of the data were not constant in Case 1 and Case 2, which represented different battery voltage conditions. It was speculated that the changes in the measured data occurred according to different states of the system, such as battery voltage and the starting position of the USV. Considering these phenomena, it was determined that using a fault diagnosis method whereby each condition was classified based on simple rules, such as defining thresholds, was difficult and not appropriate in this study. Therefore, we employed a new fault diagnosis algorithm based on PCA and entropy to perform fault diagnosis with a consideration of the correlation of selected fault features.

## 4. Methods

### 4.1. PCA

PCA is one of the statistical techniques for reducing data dimensionality while preserving the structure and characteristics of multiple data. This technique has the advantage of reducing the dimensionality of multiple data and visualizing results in a graph, allowing the visual evaluation of the similarities and differences among data and the possibility

of grouping them [30,31]. The equation for singular value decomposition, which was utilized as a representative technique for obtaining the principal component, is shown in Equation (7).

$$X_{[i \times j]} = U_{[i \times k]} S_{[k \times k]} V_{[j \times k]}^T \tag{7}$$

where *U* and *V* are orthogonal matrices, *S* is a diagonal matrix sorted in order of big singular values and *X* denotes a matrix containing existing data. Additionally, the principal axes are represented by the column vector of *V*.

$$Z_{[i \times k]} = U_{[i \times k]} S_{[k \times k]} \tag{8}$$

In Equation (8), *Z* represents the principal component score of each principal component. These scores are acquired by directly projecting existing data onto each principal component axis with column vectors of *Z*, which correspond to original data expressed at the principal component dimension.

In addition to PCA, there have been studies on algorithms, such as t-SNE and Uniform Manifold Approximation and Projection, which are widely used visualization methods for dimensionality reduction. However, these methods require longer computation time for increasing numbers of data. Furthermore, there are various hyperparameters to set for the application of these algorithms, which makes the analysis of results according to parameter tuning difficult. Therefore, in this study, the PCA technique, which is relatively simple and poses less difficulty in analyzing results, was employed.

### 4.2. Shannon Entropy

Entropy is a statistical concept first introduced by Shannon in 1949. It allows the quantification of data uncertainty of random variables, and the equation for entropy is as shown in Equation (9).

$$H(Y) = H(P_1, \cdots P_n) = -\sum_{i=1}^{n} P_i log_2 P_i \tag{9}$$

where $H(Y)$, *Y* and $P_i$ denote the information entropy, a random variable and the *i*-th probability, respectively. An occurrence with high probability can be considered a natural event; thus, little information is gained from the observation. In contrast, significant information can be gleaned from observing an uncommon occurrence; thus, it can be considered that the amount of information and probability are inversely proportional [32]. Therefore, low and high entropy indicate concentrated and dispersed data distributions, respectively. Based on this, it was predicted that under normal conditions the entropy value will be small because of the small variation in the data; under fault conditions, on the other hand, the entropy value will increase, exhibiting a direct correlation with the severity of the fault.

Since the PCA results obtained in this study were the results of visualizations after dimension reduction to 3D or 2D, human input was required for the classification of faults. To simplify this process and to reduce the time required for fault detection and diagnosis, the PCA results were quantified via the entropy method for comparison.

### 4.3. Data Preprocessing

The fault features used in this study had different units and properties, making it difficult to perform the analysis directly. Considering these differences, the data were preprocessed before PCA was applied to them. During preprocessing, the mean was set as zero through mean centering. Then, owing to the changes in data by the fault conditions associated with each fault feature, scaling was conducted so that the symptoms of faults could be highlighted. As a last step, to reduce the effect of the data having different units, the values were normalized to a range between 0 and 1.

In the preprocessing, the degree of scaling was set as a parameter, such that when the results after PCA were visualized, the symptoms of faults could be visually distinguished. The details are explained with figures in Section 5. The scaling parameter values were derived through a process of trial and error.

## 5. Analysis of the Fault Diagnosis Algorithm

### 5.1. Fault Diagnosis Algorithm

In this study, faults were diagnosed by classifying the breakage of the USV thruster blade and entanglement of floating object conditions based on the correlations between data. To this end, PCA, whereby the structure and characteristics of data are preserved with minimal loss and dimensionality is reduced, was performed for the visual representation of each fault condition. Furthermore, to simplify the process of fault detection and diagnosis, the entropy method was employed to quantify the visualized PCA results. The proposed fault diagnosis algorithm developed using PCA and entropy is illustrated in Figure 8.

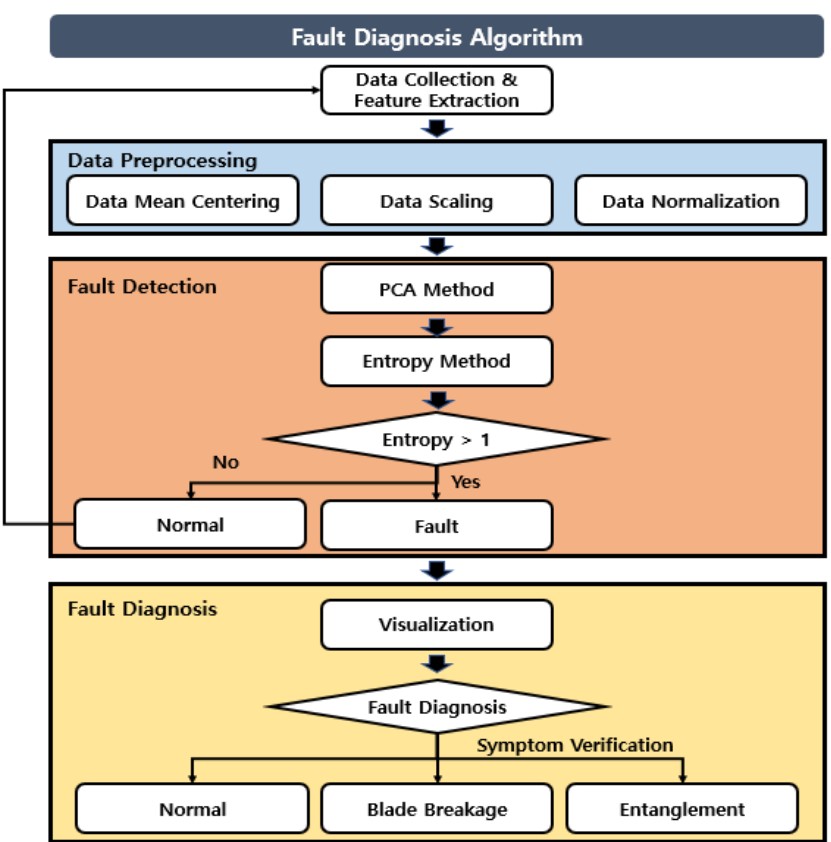

**Figure 8.** Fault diagnosis algorithm based on PCA and entropy.

The fault diagnosis algorithm was composed of the following processes: data acquisition, preprocessing, fault detection and fault diagnosis. First, the fault feature data were acquired, then preprocessed. The dimensionality of the data was reduced by applying the PCA technique in the fault detection process. Subsequently, the value was quantified through entropy. The entropy value was compared to the pre-selected threshold, and if it was smaller, it was considered normal. A larger value indicated a fault. When a fault was detected, the algorithm proceeded to the fault diagnosis process, visualized the PCA result and analyzed the symptoms of each fault to perform fault diagnosis.

### 5.2. PCA and Entropy-Based Fault Detection

By comparing the changes in the data under the fault conditions obtained from the water-tank experiments described in Section 3, it was confirmed that values for vibration

and current were sensitive to fault occurrence. The magnitude of vibration increased under the fault conditions, and especially when floating objects were entangled, the values of current data increased. Furthermore, when the objects were entangled in a thick rope or net, the changes in the values of vibration and current were even more significant. Therefore, in this study, the PCA results were orthogonally projected onto the plane of the vibration axis and the current axis. Then, the entropy values were calculated for the planes with orthogonal projection.

Table 3 presents a summary of the ranges of the entropy values in the vibration and current planes and the mean values for each condition, respectively, for Case 1 and Case 2, the conditions of different battery voltage values. The mean values of the entropy for the results of a normal blade were 0.9035 and 0.8837, which are smaller than 1. Therefore, in this study, the threshold value was set to 1; the entropy value was defined as normal or abnormal, depending on whether it fell below or above 1, to perform fault detection.

**Table 3.** Ranges of values and mean values of entropy for the planes of vibration and current for each condition.

|  | Case 1 | | Case 2 | |
|---|---|---|---|---|
|  | **Range** | **Average** | **Range** | **Average** |
| Normal | 0.8649~0.9301 | 0.9035 | 0.8484~0.8991 | 0.8837 |
| 1 cm breakage | 1.6153~1.3526 | 1.3355 | 1.3078~1.3385 | 1.3263 |
| 2 cm breakage | 1.4818~1.5010 | 1.4908 | 1.4822~1.5334 | 1.5204 |
| 3 cm breakage | 1.3711~1.4173 | 1.3861 | 1.3349~1.3752 | 1.3542 |
| Thin rope | 1.4682~1.5381 | 1.5028 | 1.2010~1.3204 | 1.2718 |
| Thick rope | 1.8853~2.0438 | 1.9488 | 2.0706~2.0794 | 2.0758 |
| Net | 1.7219~2.0934 | 1.9448 | 1.6069~1.6595 | 1.6379 |

The vibration and current values changed more significantly in the case of entanglement in the thick rope and the net compared to the other fault conditions. The entropy values also reflected these characteristics. From Table 3, it can be observed that the entropy values for the thick rope and the net in the entanglement conditions were greater than 1.6. Therefore, in this study, the thick-rope and net entanglement conditions were classified as conditions with relatively large variations in data owing to fault occurrence, whereas the breakage conditions and the thin-rope entanglement condition, with entropy values between 1 and 1.6, were classified as conditions with relatively small changes in data.

*5.3. Fault Diagnosis through Visualization*

Figure 9 shows the visualized results obtained after reducing the dimensionality of the data acquired under normal conditions to 3D through PCA. Each blue dot indicates existing data expressed in the reduced principal component dimension. The axes represent each principal component and contain many features of the data in the order of the principal components. The data expressed in the text indicate that the fault feature vectors had significant influence on each principal component axis.

In this study, considering the characteristics of the data and the changes in them under the fault condition analyzed in Section 3, the types of vectors with significant influences on each principal component were adjusted such that they became the fault features. Through this adjustment, it was possible to perform visual classification by examining the symptoms of the faults using the types of fault feature vectors with significant influence on the principal components. In the data preprocessing process, scaling parameters were adjusted to highlight the feature factors to enable visual classification, and the scaling parameter values applicable to all conditions were derived through trial and error.

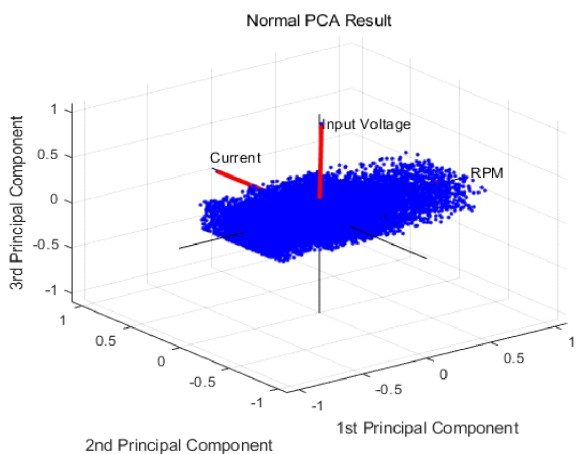

**Figure 9.** Visualized 3D PCA result of the normal condition.

From the normal condition in Figure 9, it can be observed that the RPM vector had the most significant influence on the first principal component axis, and the consumed current and input voltage had a significant influence on the next principal component axes. The effect of input voltage had little correlation with the fault conditions; thus, the intention of the visualization was to enable clear visual distinction of the normal condition. For the rest of the results, the types of vectors with significant influence on each principal component axis were visually expressed, and the types of fault feature vectors with significant influence on each principal component axis are summarized in Table 4.

**Table 4.** Types of vectors with significant influence on each principal component under different conditions.

|  | **1st PC Axis** | **2nd PC Axis** | **3rd PC Axis** |
|---|---|---|---|
| Normal | RPM | Current | Input voltage |
| 1 cm breakage | RPM | Current | Vibration |
| 2 cm breakage | RPM | Vibration | Current |
| 3 cm breakage | RPM | Current | Vibration |
| Thin rope | Current | RPM | Vibration |
| Thick rope | Vibration | Current | Input voltage/RPM |
| Net | Current | Vibration | Input voltage/RPM |

Upon detection, a fault was classified according to whether the change in data was significant or not under the fault condition based on the entropy value. The entropy value was larger than 1.6 under the thick-rope and net conditions, fault conditions under which the vibration and current data changed significantly, as shown in Figure 10.

Considering relatively significantly changes in the data, the visualization shows features appearing on the first and second principal component axes. In addition, as shown in Table 2, in the case of the thick rope, owing to greater sensitivity to vibration than to the current, it can be observed that vibration had a more significant influence on the first principal component axis. In the case of the net, owing to greater sensitivity to the consumed current than to vibration, the current had a more significant influence on the first principal component axis.

The results in Figure 11 represent cases where changes in data were relatively small. With reference to Table 4, these results show that, similar to the normal condition, the effects of RPM and current on the principal component axes were significant. Further, unlike the results for the normal condition, the effect on the principal component owing to the increase in vibration was evident. In breakage conditions (a), (b) and (c), the influence of RPM on the first principal component was significant, and the effects of vibration and current were apparent on the second and third principal component axes. In the (d) condition, which

was the thin-rope entanglement, the influence of current was observed on the first principal component axis owing to the influence of the consumed current.

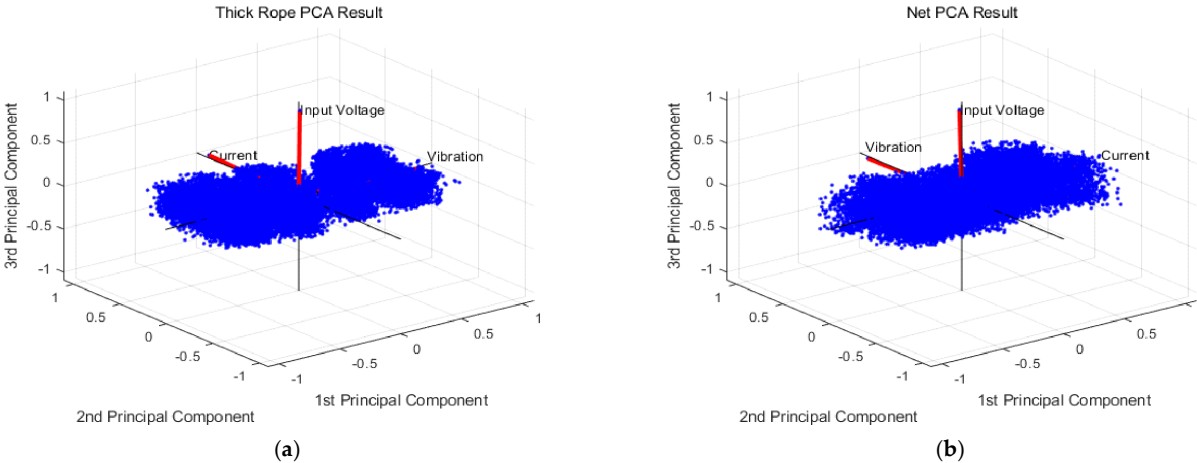

**Figure 10.** Visualized 3D PCA results for fault conditions associated significant data changes. (**a**) Thick-rope entanglement condition; (**b**) Net entanglement condition.

**Figure 11.** Visualized 3D PCA results for cases with small changes in data under fault conditions. (**a**) One-centimeter breakage condition; (**b**) Two-centimeter breakage condition; (**c**) Three-centimeter breakage condition; (**d**) Thin-rope entanglement condition.

For these conditions, by examining the vibration and current planes, the faults could be diagnosed by classifying each condition. The planes of each condition are illustrated in Figure 12.

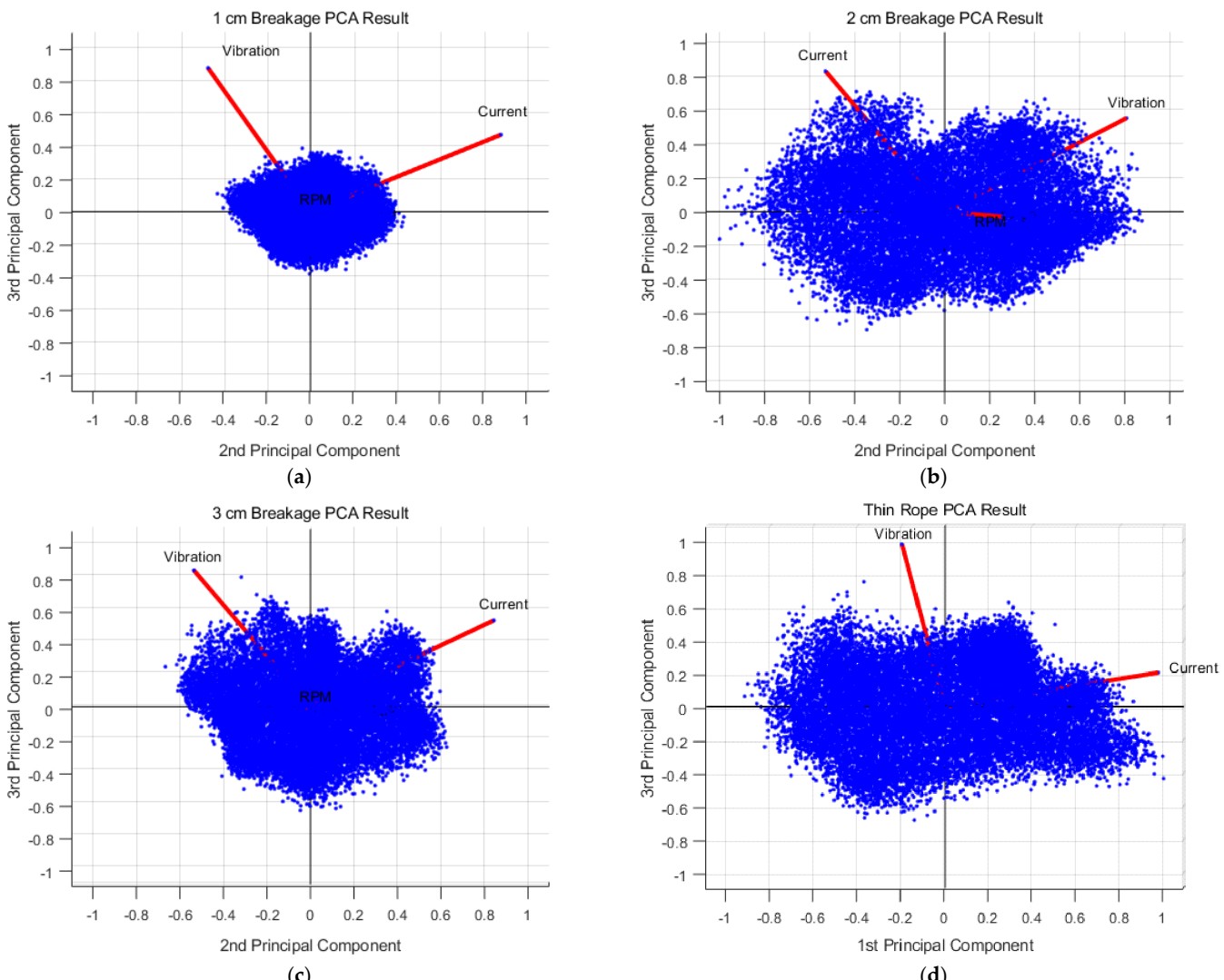

**Figure 12.** Visualized 2D PCA results for cases with small changes in data under fault conditions (vibration and current planes). (**a**) One-centimeter breakage condition; (**b**) Two-centimeter breakage condition; (**c**) Three-centimeter breakage condition; (**d**) Thin-rope entanglement condition.

Figure 12 shows the 1 cm breakage (a), 2 cm breakage (b), 3 cm breakage (c) and thin-rope entanglement (d) conditions, respectively, and shows the data distributions for the planes of vibration and current. In the breakage conditions (a), (b) and (c), it can be observed that the data distributions formed a circular shape, which is the first characteristic of breakage conditions. Furthermore, it can be observed that the circular shape disappeared with the increasing severity of breakage. In the entanglement condition, it can be observed that the distribution exhibited an angular shape in the form of a rectangle.

Under the breakage condition, the data showed more sensitive changes for vibration than for the consumed current, which, upon analysis, was found to be because vibration tended to oscillate around an equilibrium point either periodically or randomly, thus forming a circular shape in the data distribution. In addition, the current data were examined under the breakage condition in Table 2, and it was found that the consumed current decreased with increasing length of breakage. This was found to be because the consumed current decreased owing to a decrease in the diameter and area pushing away

the water; because this characteristic was very pronounced, with the increasing severity of breakage, the influence of current increased, leading to loss of the circular distribution of data. In the case of thin-rope entanglement, the consumed current data revealed a sensitive change, as a result of which the data distribution was angular in shape.

Therefore, for the cases with relatively small changes in data under the fault conditions, the shapes in the planes of the vibration and current could be examined, and if the distribution was circular, the fault could be classified as breakage, and as thin-rope entanglement if it was rectangular.

## 6. Conclusions

In this study, a new fault diagnosis technique which considers correlations between the features of fault conditions of blade breakage and entanglement with floating objects was proposed and analyzed. The proposed fault diagnosis algorithm was developed using PCA and entropy methods. The PCA technique was applied for the visualization of data with reduced dimensionality while preserving the structure and characteristics of the data. To accelerate the diagnostic process and make it more convenient, entropy was used. To diagnose the selected faults, actual data were obtained using a USV with the underwater thruster attached. The thruster's vibration, current, RPM and input voltage data were chosen as fault features based on a series of formulas and inferences. Through a number of experiments, the viability of the chosen features was verified, and the vibration and current were observed to be more fault-sensitive than other features. Fault detection was performed using the results and entropy values in Tables 2 and 3. The entropy value in the normal condition was pre-determined as 1, and the faults were detected by comparing it with the entropy value in each fault condition. In this study, the thresholds were temporarily defined, such that the values could be changed by obtaining additional datasets.

Entropy values were also used to categorize the faults and evaluate whether the data changes were large (thick-rope and net entanglements) or small (other conditions). The vibration and current change were significant with relatively large changes in data. Based on this analysis, the scaling parameter in this study was adjusted to reduce the impact of RPM. Therefore, vibration and current were located in PC 1 and PC 2. On the other hand, when the data change was relatively minimal, the adjusted scaling parameter increased the influence of RPM. Hence, RPM could be found in PC 1 in numerous instances. Nevertheless, in order to concentrate on the analysis of current and vibration, the data were projected on a plane of these axes. On this plane, the data distribution was diagnosed to classify other conditions (breakages and thin-rope entanglement). Through this proposed fault diagnosis algorithm, it was possible to classify the fault conditions of thruster blade breakage and entanglement of floating objects. Furthermore, the possibility of fault diagnosis based on fault severity was also confirmed, thus verifying the performance of the proposed algorithm.

As analyzed using the actual USV fault experimental results in Section 3.3, the values for each fault feature were inconsistent among trials. It was assumed that the causes were nonlinearity of the USV, environmental disturbance and sensor noise. The correlation between fault features was examined to overcome this limitation, and the results of multiple data were visualized for accessible analysis. In addition, for a more intuitive classification, the faults were diagnosed by checking the types of features influenced by each principal component axis and the data distributions. Unfortunately, in this study, the fault diagnosis of breakage and entanglement for a USV thruster with visual examination required a human's subjective evaluation. This resulted from the different fault conditions being categorized using the three axes of 3D space. Nevertheless, the findings of this study show the potential for the diagnosis of the severity of faults as well as the types of faults. More intuitive classifications can be anticipated if the number of cases that must be categorized can be decreased. In addition, if the scaling parameters can be properly identified during data preprocessing, this algorithm can be applied to other systems, as well as in environments significantly affected by disturbances. Furthermore, although automatic fault detection

was demonstrated by quantification using entropy, human evaluation was still required for fault diagnosis. Therefore, a follow-up study on fully automatic fault diagnosis is needed.

**Author Contributions:** Conceptualization, H.-S.C. and K.-B.C.; methodology, K.-B.C.; software, K.-B.C.; validation, K.-B.C., H.C., J.-H.P., J.H., D.J., J.L., S.-K.J., J.Y. and J.C.; formal analysis, K.-B.C. and H.C.; investigation, K.-B.C. and H.C.; resources, K.-B.C. and H.-S.C.; data curation, K.-B.C.; writing—original draft preparation, K.-B.C.; writing—review and editing, K.-B.C., H.C., J.-H.P., J.H., D.J., J.L., S.-K.J., J.Y. and J.C.; supervision, H.-S.C.; project administration, H.-S.C.; funding acquisition, H.-S.C. All authors have read and agreed to the published version of the manuscript.

**Funding:** This work was supported by the Unmanned Vehicles Core Technology Research and Development Program through the National Research Foundation of Korea (NRF) and the Unmanned Vehicle Advanced Research Center (UVARC), funded by the Ministry of Science and ICT, the Republic of Korea (NRF-2020M3C1C1A02086321).

**Institutional Review Board Statement:** Not applicable.

**Informed Consent Statement:** Not applicable.

**Data Availability Statement:** Not applicable.

**Acknowledgments:** The authors acknowledge all members of the Korea Maritime & Ocean University Intelligent Robot & Automation Lab.

**Conflicts of Interest:** The authors declare no conflict of interest.

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
