# Peer review of "A Research on Fault Diagnosis of a USV Thruster Based on PCA and Entropy"

_applsci, doi:10.3390/app13053344_

Round 1
Reviewer 1 Report
In general, the framework of this article is reasonable. The research highlights are 1. PCA and entropy based fault detection, 2. Fault diagnosis through visualization, but the following questions still need to be answered.
1.In the introduction, the contribution of this paper should be emphasized. At present, the author does not highlight the innovation of the article
2.The introduction is too redundant, and part of the literature should be placed in the literature review chapter. At present, Lack of literature revie
3. At present, only 25 articles are seriously insufficient. The author did not sort out and review the research in this field.
4. In addition, the articles about deep learning fault diagnosis methods are recommended to the author.
https://doi.org/10.3390/ijgi10100653
https://doi.org/10.1002/qre.2760
5. The author proposed"The entropy value was compared to the pre-selected threshold, and if it was smaller, it was considered normal. A larger value indicated a fault."
It is important to determine the accuracy of the algorithm and list the literature basis
6. Need more detailed algorithm formula, not just principle introduction
7.The author proposed"in this study, the threshold value was set to 1;" It is necessary to list literature support the setting of threshold. If there is no relevant literature basis, it is necessary to prove the scientificity of selecting this threshold through experiments
8.The conclusion is not good enough. It should be discussed in combination with the result.
9. This article is biased towards application and lacks theoretical contributions. Suggest the author to add it
Author Response
Dear Reviewer:
I deeply appreciate your comments. Here I attached the response to the reviewer's comments. If you need more information, please let me know. Thank you very much for your kind reconsidertaion, and we look forward to hearing from you soon.
Sincerly yours,
Ki-Beom Choo

Reviewer 2 Report
Dear authors,
The article deals with an essential issue of fault detection on an example of the USV Thruster. The article is well-formed, abstract, introduction and conclusions are sufficient. However, I have several questions and suggestions which should be addressed:
1. References: you provided a list of 25 references. I have two suggestions:
- there is not even a single reference from the MDPI Applied Sciences, while it is a journal you would like to publish in. You should better justify your subject as suitable for the journal.
- in my opinion, you should avoid references to google search results (no. 7, 19, 21, 24). Instead, you may provide a direct link to the publication. E.g. publication 24 can be found here:
https://www.nature.com/articles/nbt0308-303
2. Measuring Equipment: page 6. NI-9215 is a four analogue inputs module, while NI-9230 and NI-9234 are 3-channel and 4-channel sound & vibration input modules. Could you describe in more detail how you used NI-9215 to measure the thruster current and NI-9230 to measure the thruster input voltage (Table 1)?
3. Methodology: you identified different parameters to define vectors with significant influence on each component under different conditions (Table 4, e.g. RPM, Current and Vibration for a 1cm breakage).
- I'm interested in how you assign the three most significant parameters during operation. Are these the three whose values deviate the most from the regular operation?
- In the case of Thick rope and Net, you have two parameters in the 3rd PC axis. How do you deal with it in practice?
- Comparison of Fig.7 with Table 5. Fig.7 shows a shallow change of the Current and RPM in the cases of 1cm, 2cm and 3cm breakage. However, in Table 5, RPM is identified as the first significant parameter (1st PC axis), and Current as the 2nd or 3rd. Can you comment on this?
4. Editorial: In Fig.5, there are very slight variations in the photos showing breakages. Maybe the relevant areas of the thruster blade could be enlarged.
5. Discussion: could you discuss using an alternative way to identify faults, e.g., a neural network, a fuzzy logic-based inference or a rule-based expert system?
Best Regards,
Author Response

(The authors gave the same response as above.)

Round 2
Reviewer 1 Report
Many questions seem to be answered by the author, but the actual revision in the text is limited
Author Response
Dear reviewer
We appreciate your valuable time and consideration.
Here I uploaded the detailed response to the reviewers’ comments.
I understood that the revision of the manuscript was important among your comments, so I revised the contents intensively (Response 1-3, 5-8).
The removed contents were marked in green and the added contents were marked in yellow. In addition, I added memos to the manuscript and response files for explanation.
If you need more information, please let me know. Thank you very much for your kind reconsideration, and we look forward to hearing from you soon.
Sincerely yours,
Ki-Beom Choo

Reviewer 2 Report
Dear authors,
You made the suggested amendments. My recommendation is to accept the article.
Author Response
Dear reviewer
We appreciate your valuable time and consideration.
Thank you very much for your kind reconsideration.
Sincerely yours,
Ki-Beom Choo
Round 3
Reviewer 1 Report
The article has been carefully revised, and the author's work is worthy of recognition